# Comparative Meropenem Pharmacodynamics and Emergence of Resistance against Carbapenem-Susceptible Non-Carbapenemase-Producing and Carbapenemase-Producing Enterobacterales: A Pharmacodynamic Study in a Hollow-Fiber Infection Model

**DOI:** 10.3390/antibiotics12121717

**Published:** 2023-12-12

**Authors:** Maria V. Golikova, Kamilla N. Alieva, Elena N. Strukova, Daria A. Kondratieva, Nika F. Petrova, Mayya A. Petrova, Stephen H. Zinner

**Affiliations:** 1Department of Pharmacokinetics & Pharmacodynamics, Gause Institute of New Antibiotics, 11 Bolshaya Pirogovskaya Street, 119021 Moscow, Russia; qvimqwem@yandex.ru (K.N.A.); strukovagause@gmail.com (E.N.S.); goawaymrway@gmail.com (D.A.K.); 2National Research Centre “Kurchatov Institute”, 123182 Moscow, Russia; nikafpetrova@gmail.com (N.F.P.); petrova@img.ras.ru (M.A.P.); 3Harvard Medical School, Department of Medicine, Mount Auburn Hospital, 330 Mount Auburn Street., Cambridge, MA 02138, USA; szinner@mah.harvard.edu

**Keywords:** horizontal gene transfer, plasmids, conjugation, β-lactams, antibiotic resistance, inoculum effect, in vitro hollow-fiber dynamic model, meropenem, *Enterobacterales*

## Abstract

Resistance to carbapenems has become a problem due to *Klebsiella pneumoniae* (*K. pneumoniae*), harboring carbapenemases. Among them, there are isolates that are recognized as carbapenem-susceptible; however, these carbapenemase-producing strains with low meropenem minimal inhibitory concentrations (MICs) may pose a threat to public health. We aimed to investigate the impact of the ability to produce carbapenemases by a bacterial isolate on the effectiveness of meropenem in the hollow-fiber infection model. *K. pneumoniae* and *Escherichia coli* (*E. coli*) strains with equal meropenem MICs but differing in their ability to produce carbapenemases were used in pharmacodynamic simulations with meropenem. In addition to standard MIC determination, we assessed the MICs against tested strains at high inoculum density to test if the inoculum effect occurs. According to pharmacodynamic data, the carbapenemase-producing strains were characterized with a relatively decreased meropenem effectiveness compared to non-producers. Meanwhile, the effect of meropenem perfectly correlated with the meropenem exposure expressed as the DOSE/MIC ratio when high-inoculum (HI) MICs but not standard-inoculum (SI) MICs were used for regression analysis. It could be concluded that meropenem-susceptible carbapenemase-producing strains may not respond to meropenem therapy; the antibiotic inoculum effect (IE) may have a prognostic value to reveal the meropenem-susceptible *Enterobacterales* that harbor carbapenemase genes.

## 1. Introduction

Beta-lactam antibiotics are among the most commonly prescribed antimicrobials for a wide variety of clinical indications. Among them, carbapenems are effective against many organisms and are typically used to treat complicated infections. Unfortunately, over the last decade, resistance to carbapenems among Gram-negative bacteria has become a major problem, especially carbapenemase-producing *K. pneumoniae* [1,2,3]. These organisms can spread easily among hospitalized patients and can transmit carbapenemase genes located on different mobile genetic elements (plasmids, transposons, and insertion sequences) to other Gram-negative bacteria by horizontal gene transfer [4,5,6]. The most common carbapenemase genes are *bla*_KPC-2_ and *bla*_OXA-48_ [7,8]. Among carbapenemase producers circulating in hospitals, isolates characterized by carbapenem MICs up to the 2 mg/L susceptibility breakpoint are recognized as carbapenem-susceptible by European Committee on Antimicrobial Susceptibility Testing [9] and Clinical & Laboratory Standards Institute [10] guidelines. EUCAST guidelines do not consider whether or not the key pathogen is able to produce carbapenemases; if a given isolate is categorized as susceptible but produces carbapenemases, “it still should be reported as tested”. It is not clear if carbapenemase-producing strains with low meropenem MICs actually pose a threat to public health. As far as we know, the studies aimed at these issues are lacking. Previously, our group conducted a pharmacodynamic study in a hollow-fiber infection model (HFIM) with meropenem and carbapenemase-producing *K. pneumoniae* strains with a wide range of meropenem susceptibility, including strains with MICs below the 2 mg/L breakpoint [11]. As a result, these susceptible strains displayed intensive regrowth and decreased meropenem susceptibility during meropenem exposure, despite simulating the maximal dose of 2 g over a 3 h infusion.

In the current study, we aimed to investigate the impact of carbapenemase production by bacterial isolates with meropenem susceptibility or intermediate susceptibility, on the effectiveness of meropenem in the HFIM. The HFIM is an effective tool to study antimicrobial efficacy and has many applications, including studying the emergence of bacterial resistance and estimating the clinical efficacy of antimicrobial drugs [12,13]. To address this goal, well-characterized parental carbapenemase non-producing *K. pneumoniae* ATCC 700603 and *E. coli* C600 strains with extremely low meropenem MICs were used as recipients in mating experiments. The clinical isolates containing plasmids carrying the *bla*_OXA-48_ and *bla*_KPC-2_ genes of carbapenemases from D and A classes were used as donors. The resulting transconjugants were able to produce carbapenemases but retained meropenem MICs below the resistance breakpoint. In addition, we used the same parental *K. pneumoniae* ATCC 700603 and *E. coli* C600 strains to obtain less meropenem-susceptible variants with non-carbapenemase-producing resistance mechanisms through serial passage on meropenem-containing media. Thus, we obtained derivative strains with equal meropenem MICs but different resistance mechanisms that were used as model strains in pharmacodynamic simulations with meropenem. This allowed us to compare the influence of carbapenemase production by bacterial strains with comparable meropenem susceptibility on meropenem pharmacodynamics and the emergence of resistance. In addition to standard MIC determinations, we assessed meropenem activity against strains at high inoculum density to test if the inoculum effect is present [14]. It is well known that beta-lactam antibiotics are frequently prone to the IE if a bacterial strain produces beta-lactamase. In this study, peritoneal fluid (PF) meropenem pharmacokinetics [15] was simulated following the administration of 0.5 g every 8 h in a 0.5 h infusion during 5-day treatments in an HFIM.

The investigation of issues, referring to the role of carbapenemase production by the meropenem-susceptible strains in their ability to resist meropenem and worse treatment outcome, determines the clinical significance of the current study.

## 2. Results

### 2.1. Transconjugant and Mutant K. pneumoniae and E. coli Strains, Meropenem Susceptibility at Standard and High Inocula

The flowchart depicted in Figure 1 displays all manipulations followed to obtain model bacterial strains for the pharmacodynamic study. In the first step, we obtained transconjugants of the *K. pneumoniae* ATCC 700603 and *E. coli* C600 strains carrying plasmids containing two different carbapenemase genes. After mating of the recipient strain *E. coli* C600 with the donor strains *K. pneumoniae* 485 or 565, the respective *E. coli* tcC600/485 and tcC600/565 transconjugants were obtained. Meropenem MICs for these strains were 0.5 and 1 mg/L, respectively. After mating of the recipient strain *K. pneumoniae* ATCC 700603 and the donor strains *K. pneumoniae* 485 or 565, respective *K. pneumoniae* tc700603/485 and tc700603/565 transconjugants were obtained. Meropenem MICs for these strains were 0.5 and 4 mg/L, respectively (Figure 1).

In the second step, through passaging on meropenem-containing media, we selected mutant *K. pneumoniae* ATCC 700603 and *E. coli* C600 variants (from the parent *E. coli* C600 and *K. pneumoniae* ATCC 70060) with the same meropenem MICs as the transconjugants: (1) *E. coli* mC600/0.5 with meropenem MIC of 0.5 mg/L (the same MIC as *E. coli* tcC600/485); (2) *E. coli* mC600/1 with meropenem MIC of 1 mg/L (the same MIC as *E. coli* tcC600/565); (3) *K. pneumoniae* m700603/0.5 with meropenem MIC of 0.5 mg/L (the same MIC as *K. pneumoniae* tc700603/485); and (4) *K. pneumoniae* m700603/4 with meropenem MIC of 4 mg/L (the same MIC as *K. pneumoniae* tc700603/565).

As a result, we obtained eight bacterial strains with different meropenem MICs and the presence or absence of carbapenemase genes that were used in further pharmacodynamic simulations. Along with the meropenem MICs determined at SI, to determine if the tested bacterial strains are prone to the inoculum effect (IE), we also estimated MICs at HI. The resultant meropenem MICs for all tested strains at SI and HI are summarized in Table 1. As seen from Table 1, the susceptibility to meropenem did not depend on the inoculum density for the *E. coli* C600 and its carbapenemase-free mutants. The IE was weakly pronounced for the *K. pneumoniae* ATCC 700603 strain and its carbapenemase-free mutants. However, in the presence of carbapenemase genes, the meropenem susceptibility at HI was even more decreased. The *bla*_OXA-48_ gene affected meropenem HI MICs 2–4 times less than *bla*_KPC-2_.

### 2.2. Meropenem Pharmacodynamics with E. coli and K. pneumoniae Total and Resistant Subpopulations

At first, the pharmacodynamic experiments with all parental strains exposed to meropenem were conducted. Figure 2 shows the results of pharmacodynamic simulations with meropenem against the highly susceptible *E. coli* C600 and *K. pneumoniae* ATCC 700603 strains, as well as meropenem-resistant *K. pneumoniae* 485 and 565 strains. Not surprisingly, with the two carbapenemase-non-producing strains, the rapid and complete elimination of bacteria was observed. Conversely, carbapenemase-producers were characterized by pronounced growth and the concomitant selection of meropenem-resistant cells even though meropenem was infused into the system.

The results of simulations with transconjugant and mutant strains of *E. coli* and *K. pneumoniae* are shown in Figure 3. With both *E. coli* mC600/0.5 and tcC600/485 that have equivalent meropenem MICs of 0.5 mg/L, the antibiotic demonstrated a considerable effect, which was more pronounced with the mutant strain *E. coli* mC600/0.5; at the end of the simulations, both bacterial strains were eliminated. With *E. coli* mC600/1 and tcC600/565 (MICs of 1 mg/L), meropenem demonstrated different effects. The carbapenemase-non-producing mC600/1 was fully suppressed, while the numbers of the KPC-producing tcC600/565 increased rapidly after an initial 6 h decline; bacterial re-growth was accompanied by the intensive selection of meropenem-resistant cells (Figure 3).

Although *K. pneumoniae* m700603/0.5 and tc700603/485 had identical meropenem MICs of 0.5 mg/L, they behaved differently under antibiotic exposure. Numbers of the mutant carbapenemase-non-producing strain m700603/0.5 decreased gradually throughout the observation period, but the transconjugant OXA-48-producing strain tc700603/485 showed a modest meropenem effect and regrowth, and the selection of meropenem-resistant cells was observed after 24 h. Against *K. pneumoniae* m700603/4 and tc700603/565 with MICs of 4 mg/L, meropenem had a minimal effect, and both strains behaved similarly with rapid regrowth after an initial decline and concomitant selection of meropenem-resistant cells.

### 2.3. DOSE/MIC Relationships with the Meropenem Effect and Emergence of Resistance

To quantitatively compare the meropenem effect against bacteria with equal antibiotic MICs but different carbapenemase production, the integral parameter of the effect, ABBC, was calculated (Figure 4).

As seen in the figure, ABBCs that correspond to the mutant carbapenemase-non-producing *E. coli* and *K. pneumoniae* strains are higher than for transconjugant carbapenemase-producing strains. This difference in meropenem effect is obvious with the mC600/1–tcC600/565 pair. Acquisition by the recipient C600 strain of a plasmid carrying the KPC-carbapenemase genes dramatically affects the strain’s ability to resist meropenem.

To analyze if differences in the meropenem effect (expressed as ABBC) against strains with equal MICs but different antibiotic resistance mechanisms can be predicted using MICs determined at HI, relationships of “DOSE/MIC—ABBC” were constructed for SI (for comparison) and HI (Figure 5).

The “DOSE/MIC—ABBC” relationship is described by a sigmoid function and was stronger when HI meropenem MICs were used for the DOSE/MIC calculation compared with SI MICs: *r*^2^ 0.98 versus 0.72, respectively. Using the DOSE/MIC_SI_ index, the correlation was reduced due to the stratification of data points for carbapenemase-producing and non-producing strains. When MICs at HI were used for the analysis, this stratification was not observed.

The relationships of DOSE/MIC index with resistance were also constructed for cells at all resistance levels (from 2× to 16 × MIC) using both SI and HI MICs and were described with the Gaussian function. As an example, data for cells resistant to 4 × MIC meropenem are shown in Figure 6. The relationships for *E. coli* and *K. pneumoniae* cells resistant to 2×, 8×, and 16×MIC had similar patterns. The use of HI MICs allowed the demonstration of a stronger relationship of meropenem resistance with exposure compared with SI MICs: *r*^2^ 0.96 versus 0.54, respectively.

## 3. Discussion

The present study was designed to investigate if the effectiveness of meropenem against meropenem-susceptible *Enterobacterales* can be influenced by the presence or absence of bacterial carbapenemase genes in a certain strain. We used well-characterized parental highly meropenem-susceptible carbapenemase-free *E. coli* and *K. pneumoniae* strains to obtain pairs of model isogenic strains with equal meropenem MICs but differing in the presence of carbapenemases. To obtain carbapenemase-producing strains, horizontal gene transfer of the conjugative plasmids with carbapenemase genes was performed in mating experiments with donor clinical *K. pneumoniae* strains. As a result, we obtained four transconjugant strains, two each of *E. coli* and *K. pneumoniae*, differing by levels of meropenem susceptibility (Figure 1). Although donor *K. pneumoniae* strains were highly resistant to meropenem (MICs of 32 and 64 mg/L), after plasmid transfer to the recipient strains, the resultant plasmid-related meropenem resistance was low (meropenem MICs from 0.5 to 4 mg/L). The plasmid containing the *bla*_OXA-48_ carbapenemase gene provided meropenem MICs of 0.5 mg/L in both recipient strains, whereas the plasmid containing *bla*_KPC-2_ provided meropenem MICs of 1 and 4 mg/L in *E. coli* and *K. pneumoniae* recipient strains, respectively. The low meropenem MICs against strains with the transferred *bla*_OXA-48_ gene can be explained by the fact that the enzyme encoded by the gene has weak carbapenemase activity [15]. The KPC-carbapenemases are more powerful carbapenem-hydrolyzing enzymes [16] and provided higher meropenem MICs than the OXA-48-carbapenemase. Apparently, the high meropenem MICs of donor *K. pneumoniae* strains are due to carbapenemase production and alternative heterogeneous resistance mechanisms (i.e., reduced outer membrane permeability [17], efflux pump activation [18]).

Assuming the given meropenem MICs in transconjugants, the respective mutant carbapenemase-free strains of *E. coli* and *K. pneumoniae* were selected on meropenem-containing media. Then, all these strains were used in the comparative pharmacodynamic study in HFIM. In the HFIM, we simulated the pharmacokinetic profile of meropenem in peritoneal fluid to mimic an intraabdominal infection [15]. We chose this scenario because this infection provides optimal conditions for horizontal gene transfer during this intra-abdominal infection usually caused by *Enterobacterales* [19]. A series of kinetic time-kill curves displaying the course of both total and resistant bacterial populations were obtained for each strain. The meropenem antibacterial effect was assessed with the ABBC integral parameter; this allowed quantitative comparisons of meropenem effectiveness among related strains with equal meropenem MICs but differing by the ability of carbapenemase production. As seen in Figure 3, the ability of a strain to produce carbapenemases determines its survival capability during antibiotic exposure. This can be most clearly seen in the example of the *E. coli* transconjugant strain tcC600/565 with a meropenem MIC of 1 mg/L, which exhibited extremely intensive growth under meropenem exposure, unlike its carbapenemase-free pair strain mC600/1 that was eliminated. Therefore, it can be concluded that the specific resistance mechanism related to carbapenemase production plays a key role in the effectiveness of the antibiotic even against susceptible strains; this is not obvious with standard MIC testing. However, HI MIC testing revealed dramatic differences between strains that had equal SI MICs. All carbapenemase producers had high HI MICs and demonstrated the IE. It is worth noting that the *K. pneumoniae* ATCC 700603 strain was also characterized with the IE despite it being a non-carbapenemase producer. However, this is rather an exception as, in general, carbapenemase non-producers are rarely associated with IE [14,20]. The prognostic value of HI MICs in terms of meropenem effectiveness was confirmed with the “exposure—effect” and “exposure—resistance” relationships, which were much stronger with HI than SI MICs (Figure 5 and Figure 6).

This study provides evidence in support of the utility of detecting carbapenemase genes even in meropenem-susceptible *Enterobacterales*, especially in intra-abdominal infections. In addition, determining carbapenem MICs at high bacterial inocula to study the IE is a useful, inexpensive, and easy-to-perform option to detect organisms that harbor carbapenemases. This is especially important given that the IE is clearly linked to bacterial beta-lactamase production.

Numerous clinical studies have investigated carbapenem efficacy in infections caused by carbapenemase-producing and non-producing *Enterobacterales* [21,22,23,24,25]. However, these studies did not highlight strains with meropenem MICs below the resistance breakpoint (2–4 mg/L) that might also harbor carbapenemase genes. Attempts to distinguish carbapenemase-producers and non-producers using meropenem MICs have been made and the MIC breakpoint of 2 mg/L was proposed (i.e., strains with MICs above or below the 2 mg/L are accounted as producers or not, respectively) [26]. Unfortunately, isolates with meropenem MICs below 2 mg/L that carry carbapenemase genes are frequently found. Moreover, there is a hidden threat that initially highly susceptible non-carbapenemase-producing organisms might acquire plasmids that carry carbapenemase genes. As we have shown in this study, strains having no resistance mechanisms other than carbapenemase production may actually have meropenem MICs below the susceptibility breakpoint. Obviously, these transconjugant cells may not respond to meropenem therapy since they can produce carbapenemases.

Our study is a pilot and therefore did not include a large number of *E. coli* and *K. pneumoniae* strains with a wide range of MICs or other carbapenemase classes, for example, metallo-beta-lactamases. Additional pharmacodynamic studies with *E. coli* and *K. pneumoniae* and other clinically significant Gram-negative bacteria (e.g., *Pseudomonas aeruginosa, Acinetobacter baumannii*) are necessary to fully evaluate the role of carbapenemase-production on the antimicrobial effect of beta-lactam. Subsequent studies with a wider range of beta-lactam antibiotics would enhance the generalizability of our results.

## 4. Materials and Methods

### 4.1. Antimicrobial Agent, Bacterial Isolates, and Susceptibility Testing

Meropenem powder was purchased from Sigma-Aldrich (St. Louis, MO, USA). Two clinical isolates of *K. pneumoniae*, 485 (meropenem MIC = 32 mg/L; isolated in Saint-Petersburg in 2012 year) and 565 (meropenem MIC = 64 mg/L; isolated in Saint-Petersburg in 2011 year), carrying plasmids with *bla*_OXA-48_ (POXAAPSS2, IncL, KU159086.1) and *bla*_KPC-2_ (PKPCAPSS plasmid, IncFII, KP008371, [27]) carbapenemase genes, respectively, were used as plasmid donors in mating experiments. Strains of *E. coli* K-12 C600 and *K. pneumoniae* ATCC 700603 were used as recipients in mating experiments and as parental strains in meropenem resistance selection. Before and after each testing or experiment, carbapenemase production was verified for each bacterial strain using a modified carbapenem-inactivation method [28].

Susceptibility testing was carried out using broth microdilution techniques with a standard inoculum of approximately 5 × 10^5^ CFU/mL (SI) and a high inoculum of approximately 5 × 10^7^ CFU/mL (HI). Meropenem MICs at SI were determined according to standard recommendations using cation-supplemented Mueller–Hinton broth (CSMHB) (Becton Dickinson, Franklin Lakes, NJ, USA) [29]. Before reading, plates were incubated at 37 °C for 18–20 h. In case of determination of MICs at HI, bacterial growth in the trays was quantified by optical density at 600 nm (OD) before and after 18 h of incubation at 37 °C. MIC was taken as the minimal dilution at which the 18 h OD was equal to or less than that at time 0. Inoculum effect was defined as an eightfold or greater increase in MIC when tested with HI compared to at SI. MIC values in each case were obtained at least in triplicate, and modal MICs were estimated.

### 4.2. Mating Experiments

For mating experiments to distinguish between donor and transconjugant cells, the recipient strains of *K. pneumoniae* ATCC 700603 and *E. coli* C600 were selected on the media with rifampicin to produce rifampicin-resistant variants (rifampicin MICs of 256 mg/L). For all experiments, these rifampicin-resistant *K. pneumoniae* ATCC 700603 and *E. coli* C600 strains were used.

Bacteria were grown in Luria Broth (LB, Becton Dickinson, Franklin Lakes, NJ, USA) and Luria Agar (LA, Becton Dickinson, Franklin Lakes, NJ, USA) Media [30] at 37 °C. When required, LB Agar was supplemented with antimicrobial agents at the following final concentrations (mg/mL): meropenem (0.5–2) and rifampicin (150).

Matings were performed overnight on the LA plates. Briefly, the 1:1 mixture of the donor and recipient in the late logarithmic growth phase was plated on the LA surface and incubated at 37 °C for 18–20 h. The mixed growth was then scraped from the plate surface and then resuspended in 1 mL of saline, and to quantify the numbers of donor, recipient, and transconjugant cells, the cell mixture was diluted as appropriate and 100 µL samples were spread on appropriate selective plates with respective antibiotics. Parent strains were plated in parallel with the matings and then processed similarly to the matings as controls. Isolated colonies from matings presumed as transconjugant and parental strains from controls were used to identify recombinants and parental forms. The plasmid acquisition by recipient strains was confirmed by PCR with primers specific to genes encoding plasmid replication proteins and relaxases (positions 1039–2159 and 22713–23548 in KU159086.1 and 77542–78366 and 83825–84614 in KP008371.1, respectively). PCR was performed according to a standard protocol for amplification of fragments with a size of 1 kb [31].

### 4.3. Resistance Selection Studies under Static Conditions

To select mutants of *E. coli* and *K. pneumoniae* with lowered meropenem susceptibility (equal to the respective transconjugant MICs), a previously described serial passages technique on antibiotic-containing media was performed [32]. Briefly, parental strains (*E. coli* C600 and *K. pneumoniae* ATCC 700603) were passaged in CSMHB, containing consecutively increasing concentrations of meropenem (from 0.016 to 4 mg/L). For each subsequent passage, an inoculum was taken from the tube with the maximal meropenem concentration that showed visual growth with a turbidity equivalent to or exceeding that of a 0.3 McFarland standard. A sample was then plated on Mueller–Hinton agar (MHA, Becton Dickinson, Franklin Lakes, NJ, USA) containing the same meropenem concentration and the cycle was repeated. The incubation period of each step was 24 h. The passages in meropenem containing CSMHB were repeated 7 to 23 times, depending on the strain.

### 4.4. In Vitro Dynamic Model and Operational Procedure Used in the Pharmacodynamic Experiments

The HFIM was used to evaluate meropenem pharmacodynamics and to conduct growth control experiments. The studies were performed using a hollow-fiber bioreactor (Fresenius dialyzer, model AV400S, Fresenius Medical Care AG, Bad Homburg Germany) that represents the infection site.

The operational procedure is described in detail elsewhere [33]. Briefly, antibiotic dosing and sampling were processed automatically, using computer-assisted controls. The system was filled with sterile CSMHB and placed in an incubator at 37 °C. The inoculum of an 18 h culture of *E. coli* or *K. pneumoniae* was injected into the hollow-fiber bioreactor to produce a bacterial concentration of 10^8^ CFU/mL. After a 2 h incubation, samples were obtained to determine the starting bacterial concentration; then, the infusion of CSMHB with antibiotic was initiated. The duration of each experiment was 120 h. To verify the reliability of pharmacokinetic simulations, throughout each experiment, the bioreactor was sampled immediately after the end of infusion (0.5 h) and at the 5th hour of the dosing interval.

### 4.5. Antibiotic Dosing Regimens and Simulated Pharmacokinetic Profiles

Meropenem treatment mimicked the therapeutic dosing regimen: 0.5 g administered every 8 h, as a 0.5 h intravenous infusion. A mono-exponential profile in peritoneal fluid (PF) after thrice-daily dosing of meropenem with a half-life (t_1/2_) of 1.4 h was simulated [15] for 5 consecutive days with a total of 15 infusions. The pharmacokinetic parameter values were as follows: C_MAX_ = 24 mg/L, 24 h area under the concentration–time curve (AUC) = 160 (mg × h)/L. Before all pharmacodynamic simulations, the system was calibrated and preliminary in vitro pharmacokinetic experiments in CSMHB without bacteria were conducted. For “dose—response” relationships, the total daily dose of meropenem (in mg) that was infused into the system to simulate the desired meropenem pharmacokinetic profile was calculated.

### 4.6. Quantitation of the Antimicrobial Effect

Bacteria-containing samples from the hollow-fiber bioreactor were obtained to determine bacterial titer and antibiotic concentrations throughout the observation period. Total bacterial enumeration was performed by plating 100 µL samples onto tryptic soy agar (Becton Dickinson, Franklin Lakes, NJ, USA) plates. If necessary, the samples were serially diluted and were then plated onto tryptic soy agar. All plates were incubated at 37 °C for 24 h to quantify the growth. The lower limit of accurate detection was 2 log CFU/mL (equivalent to 10 colonies per plate).

To determine the time course of meropenem-resistant bacterial numbers, samples from the hollow-fiber bioreactor were plated on MHA plates with meropenem concentrations equal to 2×, 4×, 8×, and 16× MIC of the tested strain, incubated for 24–48 h at 37 °C, and then screened visually for growth. The lower limit of detection was 1 log CFU/mL (equivalent to at least one colony per plate).

To quantify the meropenem total antimicrobial effect, an integral parameter, determined as the area between the control growth and time-kill curves (ABBC) [11], was calculated. The interrelation of ABBC with the antibacterial effect is direct: the greater the effect, the higher the ABBC. Time courses of bacterial counts resistant to meropenem (for each meropenem resistance level, from 2× to 16× MIC) in pharmacodynamic experiments were characterized by the area under the bacterial mutant concentration–time curve (AUBC_M_) [11] determined from the beginning of treatment to 120 h. The interrelation of AUBC_M_ with the emergence of resistance is direct: the greater the resistant cells growth, the higher the AUBC_M_.

### 4.7. Statistical Analysis

The reported SI and HI MIC data were obtained by calculation of the respective modal values. In pharmacodynamic and growth control experiments, bacterial count data were calculated as arithmetic mean ± standard deviations for three replicate experiments. Based on these data, kinetic growth and time-kill curves were constructed. Assuming that the coefficient of variation for log CFU/mL data was less than 10%, to facilitate figure viewing, we did not place data point error bars, in order to not interfere with the kinetic curves.

The ABBC versus log(DOSE/MIC) curve was fitted by the sigmoid function:*Y* = *Y*_0_ + *a*/{1 + exp[−(*x* − *x*_0_)/*b*]}(1)
where *Y* is ABBC; *x* is log(DOSE/MIC); *a* are maximal values of the antimicrobial effect; *x*_0_ is x corresponding to *a*/2; *b* is a parameter reflecting sigmoidicity.

A Gaussian function was used to fit the AUBC_M_ versus DOSE/MIC data sets:*Y* = *a*/{1 + exp[−(*x* − *x*_0_)/*b*]}(2)
where *Y* is AUBC_M_; *x* is log(DOSE/MIC); *x*_0_ is log DOSE/MIC that corresponds to the maximal value of *Y*; *a* and *b* are parameters.

All calculations were performed using SigmaPlot 12 software (Systat Software Inc., headquartered in San Jose, CA, USA).

## 5. Conclusions

The present study revealed that (1) unlike carbapenemase-non-producing *E. coli* and *K. pneumoniae*, meropenem-susceptible carbapenemase-producing strains may not respond to meropenem therapy; (2) the antibiotic IE may have a prognostic value to reveal meropenem-susceptible *Enterobacterales* strains between the isolates that harbor carbapenemase genes; and (3) there is a hidden threat that bacteria devoid of antibiotic resistance mechanisms may contain carbapenemase genes even if their meropenem MICs fall below the susceptibility breakpoint.

## Figures and Tables

**Figure 1 antibiotics-12-01717-f001:**
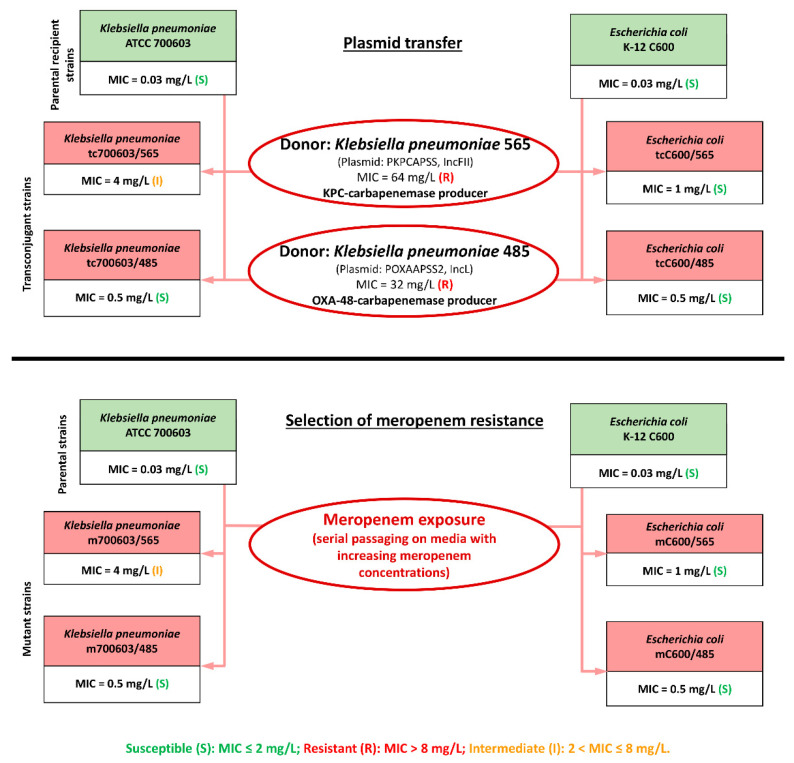
Flowchart followed to obtain bacterial strains for pharmacodynamic study.

**Figure 2 antibiotics-12-01717-f002:**
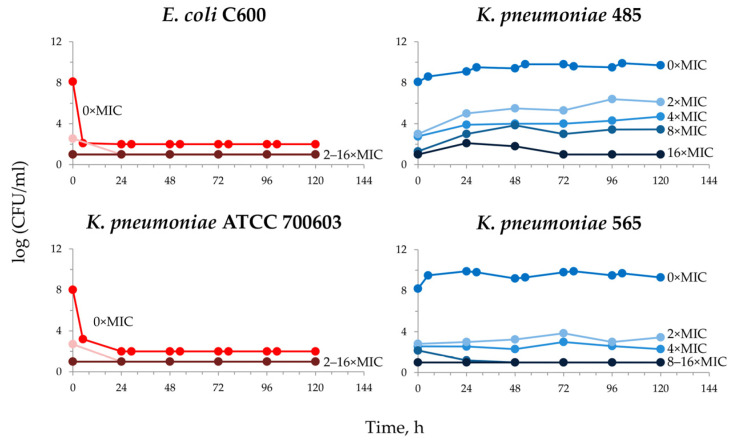
Time courses of the total bacterial population (0 × MIC) and meropenem-resistant (2×, 4×, 8× and 16 × MIC) sub-populations of parental carbapenemase-non-producing (**left** panel) and carbapenemase-producing (**right** panel) strains of *E. coli* and *K. pneumoniae* in pharmacodynamic experiments.

**Figure 3 antibiotics-12-01717-f003:**
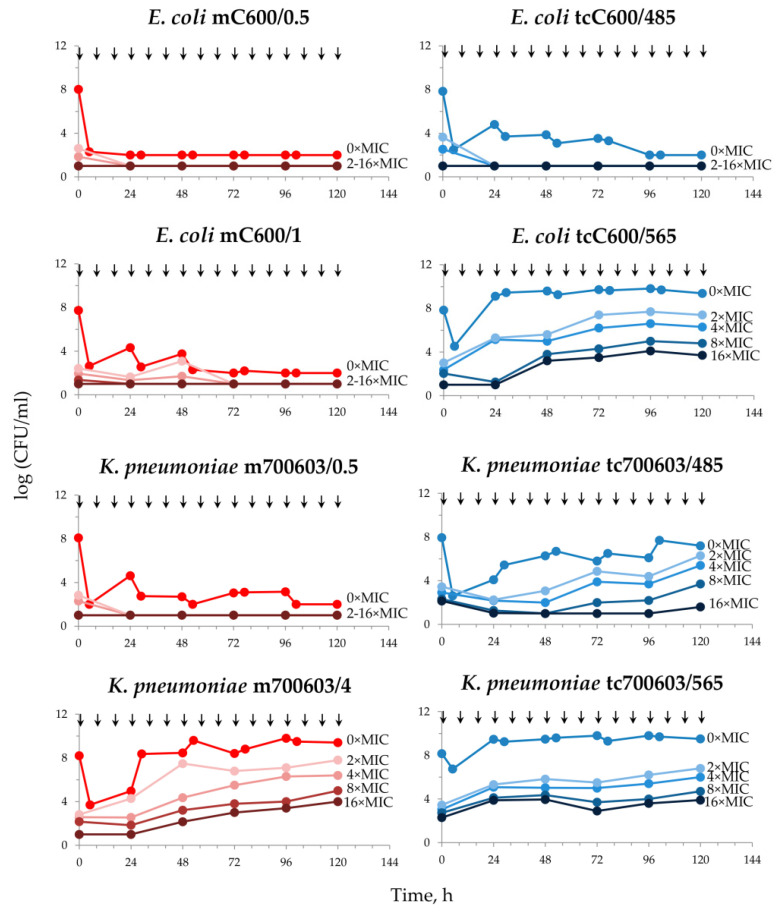
Time courses of the total bacterial population (0 × MIC) and meropenem-resistant (2×, 4×, 8× and 16 × MIC) sub-populations of mutant carbapenemase-non-producing (**left** panel) and transconjugant carbapenemase-producing (**right** panel) strains of *E. coli* and *K. pneumoniae* in pharmacodynamic experiments. Arrows indicate the start of meropenem infusion.

**Figure 4 antibiotics-12-01717-f004:**
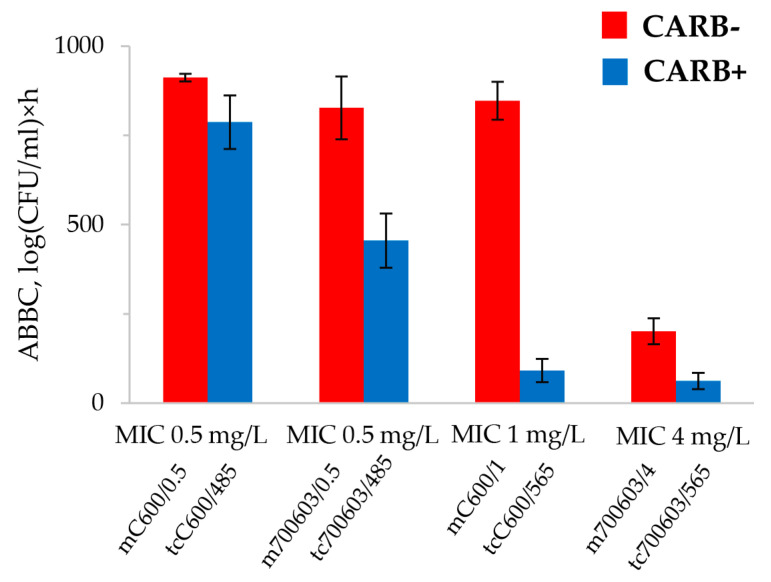
The ABBC values for mutant and transconjugant *E. coli* and *K. pneumoniae* strains grouped by meropenem susceptibility. “CARB+” indicates that the strain is a carbapenemase-producer, “CARB−” indicates that the strain is a non-carbapenemase-producer.

**Figure 5 antibiotics-12-01717-f005:**
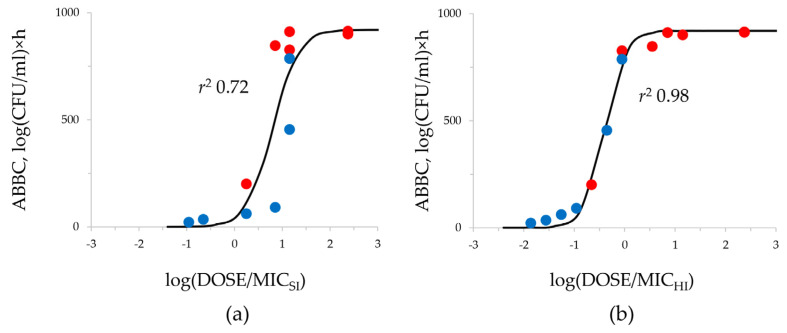
Relationship between the meropenem effectiveness and its total daily dose related to the *K. pneumoniae* MIC at SI (**a**) and at HI (**b**). Red circles—carbapenemase-non-producers; blue circles—carbapenemase-producers. DOSE/MIC relationship with ABBC fitted by Equation (1): (SI) *x*_0_ = 0.798, a = 920, b = 0.267; (HI) *x*_0_ = −0.391, a = 920, b = 0.204.

**Figure 6 antibiotics-12-01717-f006:**
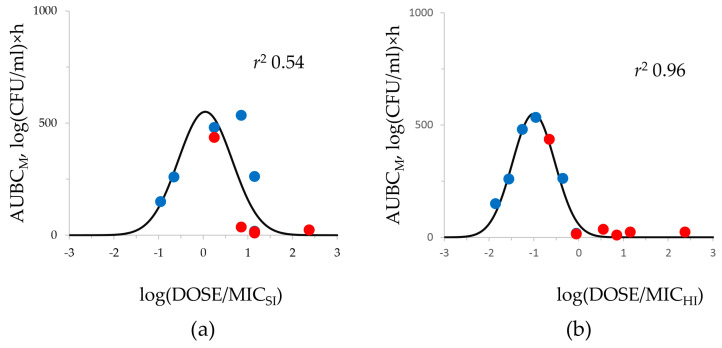
Relationship between the development of meropenem resistance (expressed as parameter AUBC_M_) of tested *E. coli* and *K. pneumoniae* strains under antibiotic exposure and meropenem total daily dose related to MIC at SI (**a**) and at HI (**b**) against respective bacterial isolate. Red circles—carbapenemase-non-producers; blue circles—carbapenemase-producers. DOSE/MIC relationship with AUBC_M_ for mutants resistant to 4 × MIC of meropenem; data fitted by Equation (2): (**a**) *x*_0_ = 0.044, a = 550, b = 0.605; (**b**) *x*_0_ = −1.004, a = 550, b = 0.474.

**Table 1 antibiotics-12-01717-t001:** Modal MICs (mg/L) of meropenem at SI and HI against *E. coli* and *K. pneumoniae* strains.

Bacterial Strain	Carbapenemase Producer	Meropenem MIC at SI, mg/L	Meropenem MIC at HI, mg/L	Fold MIC Increase	IE
C600	No	0.03	0.03	1	−
ATCC 700603	No	0.03	0.5	16 *	+
485	Yes	32	256	8 *	+
565	Yes	64	512	8 *	+
mC600/0.5	No	0.5	2	4	−
mC600/1	No	1	4	4	−
tcC600/485	Yes	0.5	8	16 *	+
tcC600/565	Yes	1	64	64 *	+
m700603/0.5	No	0.5	4	8 *	+
m700603/4	No	4	32	8 *	+
tc700603/485	Yes	0.5	8	16 *	+
tc700603/565	Yes	4	128	32 *	+

* MIC changes associated with IE. “−“-indicate the IE was not detected, “+”-indicate the IE was detected.

## Data Availability

Data are contained within the article.

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
