# Peer review of "Comparative Meropenem Pharmacodynamics and Emergence of Resistance against Carbapenem-Susceptible Non-Carbapenemase-Producing and Carbapenemase-Producing Enterobacterales: A Pharmacodynamic Study in a Hollow-Fiber Infection Model"

_antibiotics, 2023, doi:10.3390/antibiotics12121717_

Round 1
Reviewer 1 Report
Comments and Suggestions for Authors
Comments to the Author
This manuscript investigated the impact of carbapenemase production by bacterial isolates with meropenem susceptibility or intermediate susceptibility, on meropenem effectiveness in the Hollow-fiber infection model. The authors’ findings are interesting and worthy of publication after consideration of the following:
All species of bacteria & Enterobacterales: should be italicized throughout the manuscript
Line 21 “MICs” please write in full at the first mention then write the abbreviation & L 29& L 30 “HI”, “SI” & L32 “IE”
Line 23” Escherichia coli (E. coli)
Line 59: 2 grams
L 79& 81: reference is not required for the aim of the work
L311: please write the company and country of each media
L 288: please write blaKPC genotype
L323 “The plasmid acquisition by recipient strains was confirmed by PCR.” Authors should write the used primers and the condition of PCR. It is preferred to perform PCR and sequencing to confirm the acquisition of carbapenemase genes.
Comments on the Quality of English Language
Minor editing of English language required
Reviewer 2 Report
Comments and Suggestions for Authors
This paper describes the efficacy of meropenem against carbapenemase-producing and non-producing strains using a hollow fiber infection model (HFIM). The HFIM simulated the pharmacokinetic profile of meropenem in peritoneal fluid to mimic an intraabdominal infection. The results and conclusions are reasonable, but the conclusion that meropenem-sensitive carbapenemase-producing strains may not respond well to meropenem therapy is not unexpected. The discussion section could be improved to enhance the overall impact of the paper. In particular, the discussion section would benefit from addressing the following points:
1) What are the original and novel aspects of this study?
2) What are the clinical implications and consequences of the findings of this in vitro model study? Although the authors describe the utility of detecting carbapenemase genes even in meropenem-susceptible Enterobacterales, it is not realistic to perform this in all cases in clinical practice. While this paper touches upon the effects of the inoculum effect, further elaboration on the specific cases requiring carbapenemase testing would enhance clarity.
Reviewer 3 Report
Comments and Suggestions for Authors
Abstract
“we accessed the meropenem activity” authors means assessed?
HI MICs but not SI MICs the IE (Spell abbreviations outs first time in the abstract)
Comments on the Quality of English Language
Minor issues to be addressed
